# Quantum Dots-Loaded Self-Healing Gels for Versatile Fluorescent Assembly

**DOI:** 10.3390/nano12030452

**Published:** 2022-01-28

**Authors:** Chang Liu, Qing Li, Haopeng Wang, Gefei Wang, Haixia Shen

**Affiliations:** 1State Key Laboratory of Materials-Oriented Chemical Engineering, College of Chemical Engineering, Nanjing Tech University, Nanjing 210009, China; lc476564378@163.com (C.L.); liqing1128@njtech.edu.cn (Q.L.); 18842638765@163.com (H.W.); 2Research Institute of General Surgery, Jinling Hospital, Medical School of Nanjing University, Nanjing 210002, China

**Keywords:** quantum dots, nanocomposites, frontal polymerization, polymer gels, self-healing, fluorescence

## Abstract

From the perspective of applied science, methods that allow the simple construction of versatile quantum dots (QDs)-loaded gels are highly desirable. In this work, we report the self-healing assembly methods for various fluorescent QDs-loaded gels. Firstly, we employed horizontal frontal polymerization (FP) to fabricate self-healing gels within several minutes using a rapid and energy-saving means of preparation. The as-prepared gels showed pH sensitivity, satisfactory mechanical properties and excellent self-healing properties and the healing efficiency reached 90%. The integration of the QDs with the gels allowed the generation of fluorescent composites, which were successfully applied to an LED device. In addition, by using the self-healing QDs-loaded gels as building blocks, the self-healing assembly method was used to construct complex structures with different fluorescence, which could then be used for sensing and encoding. This work offers a new perspective on constructing various fluorescent assemblies by self-healing assembly, and it might stimulate the future application of self-healing gels in a self-healing assembly fashion.

## 1. Introduction

Fluorescent nanomaterials, such as quantum dots (QDs) and carbon quantum dots (CQDs) have attracted enormous attention due to their excellent photoluminescence properties [1,2]. In particular, CQDs have become one of the most popular fluorescent materials due to their stability, good water dispersibility, low cytotoxicity, better biocompatibility and accessible synthesis methods [3,4]. By virtue of these advantages, the integration of QDs with polymer gels has been widely explored [5,6,7,8,9,10,11,12,13]. Polymer gel provides an ideal matrix for hosting and immobilizing QDs, which have a unique porous structure. It should be noted that the loading of QDs not only endows the composites with fluorescence, but also other enhanced and fascinating features, such as self-healing [13]. Up to now, QDs-loaded gels have found a wide range of applications in sensors [5,6], wound dressing [7], drug delivery [8], photothermal therapy [9], optoelectronic devices [10], DNA detection [12], etc. However, these QDs-loaded gels usually show a single fluorescence signal, which hampers the precision and extension of their practical application. Thus, it is crucial to develop a feasible pathway for the construction of versatile fluorescent QDs-loaded gels.

The use of self-healing hydrogels has skyrocketed in the past few decades because of their beneficial features, which can prolong lifetimes and reduce replacement costs [13,14,15,16,17,18,19]. Up to now, a series of self-healing gels have been explored based on the dissociation and recombination of physical or chemical interactions [14]. When damaged, the physical self-healing gels can re-establish networks through the non-covalent interactions, such as hydrogen bonds, hydrophobic and host–guest interactions, etc., whereas the chemical self-healing gels usually repair the networks through the dynamic reversible covalent bonds, including imine bonds, disulfide bonds, acylhydrazone bonds and the Diels–Alder reaction. In addition, the healing abilities of these gels are mainly assessed by their morphology and mechanical properties. Digital monitoring, optical microscopy and scanning electron microscopy (SEM) can be used to track the self-healing process of scratched gels [20,21]. The recovery of mechanical properties (tensile strength, compressive strength and rheology) has been widely used, which allows the quantitative assessment of the healing efficiency, defined as the ratio of the stress of the healed and pristine gels [22]. In particular, self-healing gels are able to efficiently fuse separated materials, which endows them with excellent potential in terms of self-assembly, thus leading to enhanced properties and functions. In this case, self-healing gels serve as ideal building blocks to construct versatile structures and morphologies. For example, Harada et al. reported the self-assembly of self-healing materials through molecular recognition [23,24,25,26]. Our group also developed microfluidic-assisted assembly of self-healing gels to create ordered structures [27,28,29,30]. As an easy and effective strategy, self-healing assembly offers new opportunities for QDs-loaded gels with enhanced properties and functions. However, this approach is still in its infancy. Continued research efforts are imperative for this embryonic field, which could result in an increasing number of diverse assemblies of practical significance.

When it comes to the synthesis of smart gels, frontal polymerization (FP) is preferred as it is a rapid and efficient polymerization mode. FP is distinguished by a propagating reaction front in the local reaction zone, which travels through the whole reaction system and promotes the conversion of monomers to polymers. It should be noted that FP is driven by the enthalpy of polymerization. Once ignited, the polymerization process can be self-propagated without an external energy supply, which promises to revolutionize the future of manufacturing due to its energy-saving potential [31,32,33,34,35,36,37,38,39,40,41,42,43]. Compared to conventional batch polymerization, FP has several advantages including ease of operation, energy and time efficiency, and it is environmentally friendly [32,33]. So far, great efforts have been devoted to the development of FP; these have sparked the creation of new integrated FP techniques [34,35,36,37] and a series of functional polymers [38,39,40], especially smart gels [41,42,43].

Herein, we report the simple preparation of self-healing gels via FP within several minutes (Figure 1a,b) and quantum dots (QDs)-loaded gel assemblies for diverse applications. Monomers including acrylamide (AM), 2-acrylamido-2-methyl-1-propanesulfonic acid (AMPS) and β-cyclodextrin modified with maleic anhydride (β-CD-MAH) were chosen due to the large amount of carboxyl groups and amine groups, which generate strong hydrogen bonding interactions in the polymer network, thus endowing the gels with good self-healing properties. Considering various practical applications, we fabricated QDs-loaded gels that can be directly used as light conversion materials for LEDs. In addition, by taking advantage of the self-healing properties, we constructed QDs-loaded gel assemblies with bilayer or triangular structures with different fluorescence, and realized their application for sensing and encoding (Figure 1c). This work offers a new perspective on constructing fluorescent assemblies in a self-healing self-assembly manner, which will greatly promote the practical applications of QDs.

## 2. Materials and Methods

### 2.1. Materials

Acrylamide (AM), β-cyclodextrin (β-CD), maleic anhydride (MAH), *N*,*N*-dimethylformamide (DMF), chloroform (CHCl_3_), acetone, 2-acrylamido-2-methyl-1-propanesulfonic acid (AMPS), ethylene glycol, N, N’-methylenebisacrylamide (MBAA), ammonium persulfate (APS), *N*,*N*,*N*’,*N*’-tetramethylenediamine (TMEDA), cadmium chloride (CdCl_2_-2.5H_2_O), N-acetyl-L-cysteine (NAC), tellurium powder (Te) and sodium borohydride (NaBH4) were purchased from Sigma-Aldrich and used as received. Carbamide, zinc citrate dehydrate, sodium citrate dihydrate were purchased from Sinopharm Chemical Reagent Co. (Shanghai, China); Fe_3_O_4_ nanoparticles (20 nm) were purchased from Aladdin-reagent (Shanghai, China). Deionized water was used in all of the experiments.

### 2.2. Preparation of the CQDs

CQDs with bright green and blue fluorescence were prepared according to our previous work [3]. Briefly, Fe_3_O_4_ (10 wt%), carbamide and zinc citrate dehydrate or sodium citrate dihydrate (carbamide:citrate = 5:1 mol/mol) were added into a flask that was placed under an external magnetic field (450 kHz) for 3~5 min to excite the reaction to fabricate CQDs. The as-prepared CQDs were dispersed in water, followed by centrifugation at 10,000 rpm for 10 min to remove Fe_3_O_4_. Then the supernatant was further purified by an ultra-filtration membrane (0.22 μm). After the drying process, the solid samples were collected for further use.

### 2.3. Preparation of the CdTe QDs

CdTe QDs were synthesized using a modified procedure from a previous study [44]. First, 0.1136 g NaBH4 and 0.0638 g tellurium powder were added into a seed bottle, followed by the injection of 2 mL pure water. The seed bottle was connected with a needle to release the as-produced H_2_. After reaction (~8 h), the solution becomes clear and the NaHTe solution was obtained.

Then 0.2284 g CdCl_2_·2.5H_2_O was dissolved in 40 g water in a flask and the ligand solution (0.2448 g NAC dissolved in 5 g H_2_O) was slowly added to the flask, followed by stirring to ensure the full coordination of the ligand and Cd^2+^. The pH value was adjusted to 9 by 5 M of NaOH solution. After that, the as-prepared precursor solution was placed in an oil bath with N_2_ protection. When the temperature reached 98 °C, the NaHTe solution was rapidly injected into the system. Green-colored CdTe QDs were obtained after a 1 h reaction, which was precipitated with ethanol, they were centrifuged at 10,000 rpm for 10 min, and then dried for further use.

### 2.4. Preparation of the MAH-β-CD

Based on a previous report [45], β-CD was modified by using MAH. Briefly, β-CD (5.68 g) and MAH (4.90 g) were dissolved in 30 mL DMF in a flask. The mixture was stirred at room temperature to obtain a homogenous solution and then heated to 80 °C for 10 h. After reaction, the product was transferred to a glass beaker and precipitated with CHCl_3_ and washed thrice with acetone. Finally, the solid product was freeze-dried for more than 12 h to obtain MAH-β-CD.

### 2.5. Preparation of the Poly(AM-co-AMPS-co-MAH-β-CD) Gels via FP

A homogenous precursor was prepared with an appropriate monomer (AM, AMPS and MAH-β-CD), initiator (APS), crosslinker (MBAA) and solvent (ethylene glycol). Typically, AM/AMPS = 2:1 (*w*/*w*), MAH-β-CD = 5 wt%, ethylene glycol = 50 wt%, MBAA = 0.3 wt%, APS = 0.4 wt%, [APS]/[TMEDA] = 1:2 (*w*/*w*). The precursor was transferred into a 10 mL horizontal glass vessel, where the end was heated for 20 s with a soldering iron. The initiating temperature was 100 °C and the distance from the liquid level was 10 mm. A front formed and the polymerization could be propagated. After reaction, the as-prepared gels were taken out for further use.

### 2.6. Measurements

*Front velocity and temperature tests.* The front velocity was calculated from a frontal position–time curve, by using the slope of the curve. The thermal images and temperature were measured by a FLIR E8 IR thermal imager.

*Self-healing measurements*. To explore the self-healing properties of hydrogels, the polymer gel was cut into two pieces and reconstituted without any stimulation. The mechanical properties of the healed samples were measured with a SANS CMT6203 testing machine. By using the stress–strain curves, the healing efficiency was calculated by the mechanical strength ratio of the healed sample to the original sample. 

*Fourier-transform infrared (FTIR) spectrometry characterization*. For FTIR measurement, the sample was purified by immersing it in water for one week, and the water was changed daily. Then it was dried in a vacuum oven at 60 °C. The FTIR spectrum of the dried sample was obtained by a Thermo Nicolet-6700 Fourier Infrared spectrometer. 

*Micro-IR imaging measurements*. IR images during the self-healing process were recorded by a Thermo Scientific Nicolet iN10 infrared microscope, equipped with a liquid nitrogen cooled MCT detector (Thermo Electron Corporation, Waltham, MA, USA). The data were collected under reflection mode. IR images were captured using an aperture size of 50 μm by 50 μm. These data were analysed by OMNIC picta software (Thermo Electron Corporation, USA). 

*Rheological experiments.* The rheological properties were studied using a HAAKE MARS III instrument at 25 °C (parallel plate, 25 mm diameter).

## 3. Results and Discussions

### 3.1. Fabrication of Poly(AM-co-AMPS-co-MAH-β-CD) Gels via FP

We synthesized poly(AM-*co*-AMPS-*co*-MAH-β-CD) gels via horizontal FP. To visually observe the propagating front, methylene blue was utilized, which decomposes when heated. As clearly shown in the digital photographs (Figure 1a), the left side of the front is polymer, while the right side is monomer. The frontal position–time curve is a straight line (Figure 1b), which demonstrates that a constant front velocity was achieved. In addition, we measured the temperature profile of a fixed point during the FP process by IR thermal imaging. As shown in Figure 1c, the spreading front can be distinguished in the thermal images, where the white area (the highest temperature, *T_max_*) propagates forward with time. Figure 1d shows a typical temperature profile during the FP process. The constant horizontal stage corresponds to the region far from the front, where no polymerization occurs and the temperature remains unchanged. With the front propagating and heat diffusion, the temperature begins to rise sharply and reaches a maximum value (128 °C). Finally, the temperature declines because the front moves away. Both the stable frontal velocity and the *T_max_*, offer strong evidence for the occurrence of pure FP.

The monomer weight ratio is a the key factors that affects the frontal velocity and *T_max_* during the FP process. Here, we investigated the effect of the AM/AMPS weight ratio on the FP kinetics, as shown in Figure 1e,f. All the front position–time curves are well fitted to straight lines, verifying that the polymerization is pure FP. In Figure 1f, it can be clearly seen that when the AM/AMPS weight ratio changes from 3:2 to 8:2, the frontal velocity shows a significant upward trend, increasing from 0.97 to 1.87 cm/min. The corresponding *T_max_* increases from 125 to 143 °C. Faster frontal velocity leads to a higher reaction temperature due to less heat loss.

### 3.2. Swelling Behaviors and Mechanical Properties of Poly(AM-co-AMPS-co-MAH-β-CD) Gels

We investigated the water absorbency of poly(AM-*co*-AMPS-*co*-MAH-β-CD) gels, as shown in Figure 2a,b. It was found that the AM/AMPS weight ratio plays an important role in the swelling behaviors. When AM/AMPS = 6:2 (*w*/*w*), the equilibrium swelling ratio (ESR) of the gel reaches a maximum value of ~1525%, while a further increase or decrease in AMPS content result in a declining trend, which is ascribed to the synergy of the highly absorbent -SO_3_ and -CONH- groups, as well as the electrostatic repulsion and steric hindrance effect of AMPS [46]. In addition, the pH value also has a notable impact on the ESR, as shown in Figure 2c,d. Obviously, when the pH value of the solution is close to neutral, that is, 5~9, the ESR is high. It should be noted that the ESR gradually declines at the pH value of 9~13, while it decreases sharply in the strongly acidic range (pH value of 1~5). In the acidic range, sulfonic groups combine with H^+^ to generate sulfonic acid, reducing the complexation ability of sulfonic group with water, whereas the alkaline solution will cause repulsion between groups, which hinders the polymer chain stretching or curling, leading to the decline in the ESR. Notably, the gel shows the highest water absorbing capacity in the pH range close to that of body fluid (pH value of 6~8), which might find application in the biomedical field, such as wound dressing.

The mechanical properties of the gels are critical for their practical application; thus, we investigated the mechanical strength of the gels, as shown in Figure 2e–g. The gel shows excellent stretchability, which means it can sustain a great degree of stretching without breakage (Figure 2g). From the strain–stress curves, we can clearly see that AM/AMPS weight ratio great affects the tensile strength and elongation. In detail, the mechanical strength is proportional to the AM concentration, while the elongation is inversely proportional to the AM content (Figure 2e,f). When AM/AMPS = 8:2 (*w*/*w*), the gel possesses a highest tensile strength of 0.157 Mpa, which satisfies the requirements of practical applications.

In addition, we studied the rheological behavior of the gel samples with different AM/AMPS weight ratio. Figure 3a shows the storage modulus (G′) and loss modulus (G″) of these samples on strain sweeps at a fixed frequency of 1 rad/s. It was found that for all four groups (AM/AMPS =3:2, 4:2, 6:2, 8:2 (*w*/*w*)), the G′ values were obviously higher than the G″ values, indicating the formation of the gel network. Figure 3b demonstrates G′ and G″ values on frequency sweeps, where a frequency-dependent mechanical moduli is observed. Typically, both G′ and G″ monotonously increase with frequency. The results illustrate that the as-prepared gels possess excellent elastic properties. From the viscosity–frequency curves (Figure 3c), we can clearly see that the viscosity decreases with the frequency. The molecular interactions and molecular tangle are diminished, leading to the decline in viscosity, whereas under the same frequency, the gels exhibit different viscosity. Especially, the AM/AMPS weight ratios play a key role in the viscosity, which from high to low are 3:2, 4:2, 6:2, 8:2 *w*/*w*.

### 3.3. Self-Healing Behaviors of the Poly(AM-co-AMPS-co-MAH-β-CD)

Initially, we chose AM, AMPS and β-CD-MAH as co-monomers for the purpose of strong hydrogen bonding interactions, which endow the gels with self-healing properties. To verify our hypotheses, we cut the original gel into two parts, put them indirect contact and pressed for a moment to ensure close contact. One part of the gel was dyed by Rhodamine B to present a more visual observation of their self-healing behavior. As shown in Figure 4a, the healed sample is strong enough to sustain a great degree of stretching without breaking, indicating satisfactory self-healing ability and stretchability. We also investigated the effect of the AM/AMPS weight ratio on the healing efficiency. With the decrease in the AM content, the healing efficiency showed a strong uptrend of up to 90% (Figure 4b), which demonstrated its outstanding self-healing properties.

In theory, there are a large amount of carboxyl groups, hydroxyl groups and amine groups in the poly(AM-*co*-AMPS-*co*-MAH-β-CD) network, which can generate strong hydrogen bonding interactions, thus leading to self-healing properties, as schematically illustrated in Figure 4c. To confirm our conjecture, we performed micro-IR measurement to investigate the role of functional groups. Figure 4d,e show the FT-IR spectra and the micro-IR images of poly(AM-*co*-AMPS-*co*-MAH-β-CD) before and after healing at 3428, 3198 and 1660 cm^–1^, which correspond to the stretching vibration of hydroxyl groups, amine groups and carboxyl groups, respectively. The color variation in the IR images directly indicates the intensity change in these functional groups. Typically, blue indicates low intensity, while red represents high intensity. A large number of functional groups are exposed on the cut surface; thus, the groups’ intensity distribution is strong and correspond to the red color. As the cut gradually heals, the IR image color shows a uniform green, indicating that the distribution of these groups become uniform, which is due to the hydrogen bonding interaction between the hydroxyl groups, amine groups and carboxyl groups. These results strongly confirm our hypothesis that the hydrogen bonding interactions between functional groups endow the gels with good self-healing behaviors.

### 3.4. Applications of the Poly(AM-co-AMPS-co-MAH-β-CD) Gels in LED Devices

CQDs show broad potential in LED devices due to their excellent fluorescence stability and low toxicity [47]. Here, we doped green CQDs in P(AM-*co*-AMPS-*co*-MAH-β-CD) gels to construct fluorescent composite gel materials and applied them to green LED devices. Figure 5a,b show the LED device under power of 3.5 V, where we can clearly see that it emits bright green light. The electroluminescence (EL) spectrum of the as-prepared LED devices is shown in Figure 5c, where the emission peak of the blue LED chip is located at 410 nm and the emission peak of the CQDs-composite gel is located at 502 nm. Figure 5d shows the International Commission on Illumination (CIE) color coordinates of (0.1951, 0.4428), which correspond to green light emission. The results indicate that the CQDs-loaded fluorescent composite gels have great potential in the field of optoelectronic devices.

### 3.5. Self-Assembly of QDs-Loaded Self-Healing Gels for Fluorescent 2D Codes and Encoding

The self-healing behavior enables the construction of different structures by using the self-assembly strategy. Thus, the integration of QDs-loaded self-healing gels allows various applications including optoelectronic devices, sensing and information encryption [29,48,49]. By virtue of the self-healing assembly, we attempted to achieve double layered gels, where each layer contains different fluorescent signals, which can be utilized as a 2D code for chemical recognition. We firstly prepared self-healing gels with different fluorescent signals. Typically, two kinds of gels equipped with a single signal (Rhodamine B or QDs) were obtained, which were then put in contact to form a bilayer structure by virtue of their self-healing behavior, where the upper layer contained QDs with emission wavelengths of 525 nm, while the emission wavelengths of the bottom layer was 580 nm due to rhodamine B.

Figure 6a depicts a schematic illustration of the bilayer gel assembly with two fluorescent signals, which shows responsiveness to different organic amines. By adjusting the excitation wavelength, the fluoresce intensities of the two layers were kept almost the same, and the fluorescent spectrum in the absence of organic amines was used as a control. When the upper layer contacts with organic amines, the fluorescent signal from CdTe QDs changes and acts as the sensing signal R_1_, whereas the signal R_2_ from rhodamine B remains unchanged and serves as control. The ratio γ (γ = R_1_/R_2_) demonstrates the final sensing parameter. As shown in Figure 6b,c, after the upper layer has contact with TEOA (triethanolamine), PPA (n-propylamine), TEA (triethylamine), TMEDA and EDA (ethylenediamine), the bilayer-structure exhibits varied luminescence intensity. The corresponding γ values are trending downward; these are 0.950, 0775, 0.668, 0.443 and 0.178, respectively. The γ value serves as an indicator to translate fluorescent signal into 2D codes for the recognition of different organic amines.

To better illustrate the self-healing assembly for complex structures and functions, we also fabricated gel beads with different fluorescent signals. Rhodamine B, fluorescein and CQDs were chosen to be encapsulated into microspheres by microfluidics to obtain three types of gel beads with different emission wavelengths (Figure 6d), and then the obtained gel beads with bright red (R), green (G), and blue (B) fluorescence were employed to construct a fluorescent assembly. As indicated in Figure 6e, a combination of (R, G, B) was designed, emitting different emission wavelengths of 575 nm, 510 nm and 440 nm. These multiple fluorescence signals hold great potential in the encoding process, and could be easily decoded by fluorescence analysis. Such a method provides an effective way to generate fluorescent-coded microbeads for potential applications such as biological coding and efficient information storage.

## 4. Conclusions

In conclusion, here, we demonstrated a self-healing assembly strategy to construct various fluorescent structures. Firstly, a new self-healing P(AM-*co*-AMPS-*co*-MAH-β-CD) gel material was rapidly synthesized in a few minutes by FP, which showed excellent self-healing properties and a healing efficiency of up to 90%. The micro-IR measurement confirmed that the self-healing mechanism was based on the hydrogen bonding interactions between large amounts of functional groups (carboxyl group, hydroxyl group and amine group) in the polymer network. With regard to practical applications, we utilized QDs to endow the gel with fluorescent properties and successfully constructed LED devices. Moreover, based on its self-healing behavior, the self-assembly of QDs-loaded gels was performed to create bilayer and triangular structures with different fluorescence, where the former serves as a fluorescent 2D code to recognize different organic amines, while the latter acts as a fluorescent code for information encryption. This work provides a convenient and easy-to-perform way to construct versatile QDs-loaded gel materials by using a self-assembly strategy, which will greatly stimulate the practical application of fluorescent composites.

## Data Availability

Not applicable.

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
