# Peer review of "Quantum Dots-Loaded Self-Healing Gels for Versatile Fluorescent Assembly"

_nanomaterials, 2022, doi:10.3390/nano12030452_

Round 1
Reviewer 1 Report
The paper entitled "Quantum dots-loaded self-healing gels towards versatile fluorescent assembly" presents some new results on functional nanocomposites synthesis and characterization. The research is appropriated for the journal, however some minor comments arise. Furthermore, there are several grammar errors, English level need to be improved.
The introduction section can be improved with references on self-healing gels and related experiments to evaluate the healing performances. Tensile testing is not the only way to analyze mechanical healing performances, nor the more accurate. In this view, a paragraph in this direction can justify the selection of simple tensile tests for healing evaluation.
In figure 1, sub figures b and e seem a repetition. This figure can be compacted. Some additional experiments can help in the understanding of the different behavior of the synthetized gels. For example, measurement of transitions temperatures (DSC) or rheological measurement. The latter, will give a wider prospective on elastic and viscous response of the materials with a possible link on the healing performances. It has been observed that viscoelasticity play crucial role on the recovery of mechanical performances in healing systems, furthermore the addition of fillers should be better evaluated and linked with the observed functional response.
Author Response
Comments:
The paper entitled "Quantum dots-loaded self-healing gels towards versatile fluorescent assembly" presents some new results on functional nanocomposites synthesis and characterization. The research is appropriated for the journal, however some minor comments arise. Furthermore, there are several grammar errors, English level need to be improved.
- The introduction section can be improved with references on self-healing gels and related experiments to evaluate the healing performances. Tensile testing is not the only way to analyze mechanical healing performances, nor the more accurate. In this view, a paragraph in this direction can justify the selection of simple tensile tests for healing evaluation.
Response: Done. We have adopted the reviewer’s suggestion. Accordingly, we have added the development of self-healing gels and related experiments to evaluate the healing performances.
The modified text and figure are as follow:
Up to now, a series of self-healing gels have been explored, which were based on the dissociation and recombination of physical or chemical interactions [14]. When damaged, the physical self-healing gels can re-establish networks by the non-covalent interactions, such as hydrogen bonds, hydrophobic and host–guest interactions, etc. Whereas the chemical self-healing gels usually repair the networks through the dynamic reversible covalent bonds, including imine bonds, disulfide bonds, acylhydrazone bonds and Diels–Alder reaction. In addition, the healing ability is mainly assessed by morphology and mechanical properties. Digital monitoring, optical microscopy and scanning electron microscopy (SEM) can be used to track the self-healing process of scratched gels [20, 21]. The recovery of mechanical properties (tensile strength, compressive strength and rheology) has been widely used, which allow quantitative assessment of the healing efficiency, defining as the ratio of the stress of the healed and pristine gels [22].
- Zhang, Y.; Tao, L.; Li, S.; Wei, Y. Synthesis of Multiresponsive and Dynamic Chitosan-Based Hydrogels for Controlled Release of Bioactive Molecules. Biomacromolecules 2011, 12, 2894-2901.
- Yu, X.; Cao, X.; Chen, L.; Lan, H.; Liu, B.; Yi, T. Thixotropic and self-healing triggered reversible rheology switching in a peptide-based organogel with a cross-linked nano-ring pattern. Soft Matter 2012, 8, 3329-3334.
- Burattini, Stefano.; Greenland, Barnaby. W.; Chappell, David.; Colquhoun H. M.; Hayes, Wayne. Healable polymeric materials: a tutorial review. Chem. Soc. Rev. 2010, 39, 1973-1985.
- In figure 1, sub figures b and e seem a repetition. This figure can be compacted. Some additional experiments can help in the understanding of the different behavior of the synthetized gels. For example, measurement of transitions temperatures (DSC) or rheological measurement. The latter, will give a wider prospective on elastic and viscous response of the materials with a possible link on the healing performances. It has been observed that viscoelasticity play crucial role on the recovery of mechanical performances in healing systems, furthermore the addition of fillers should be better evaluated and linked with the observed functional response.
Response: Done. Thanks for the reviewer’s comments. We have adopted the reviewer’s suggestion. Accordingly, the rheological measurement was conducted.
The modified text and figure are as follow:
Rheological experiments. The rheological properties were studied by a HAAKE MARS III instrument at 25 °C (parallel plate, 25 mm diameter).
In addition, we studied the rheological behavior of the gel samples with different AM/AMPS weight ratio. Figure 3a shows the storage modulus (G′) and loss modulus (G″) of these samples on strain sweeps at a fixed frequency of 1 rad/s. It is found that for all the four groups (AM/AMPS =3:2, 4:2, 6:2, 8:2 (w/w)), G′ values were obviously higher than G″ values, indicating the formation of gel network. Figure 3b demonstrates G′ and G″ values on frequency sweeps, where a frequency-dependent mechanical moduli is observed. Typically, both G′ and G″ monotonously increase with frequency. The results illustrate the as-prepared gels possess excellent elastic properties. From the viscosity-frequency curves (Figure 3c), we can clearly see that the viscosity decreases with the frequency. The molecular interactions and molecular tangle are diminished, leading to the declined viscosity. Whereas under the same frequency, the gels exhibit different viscosity. Specially, the AM/AMPS weight ratios play a key role on the viscosity, which from high to low is 3:2, 4:2, 6:2, 8:2 w/w.
Figure 3. Storage modulus (G′) and loss modulus (G″) values on (a) strain sweeps and (b) frequency sweeps. (c) Viscosity-frequency curves of the gels with different AM/AMPS weight ratios.

Reviewer 2 Report
The paper under consideration reports the formation of self-healing hydrogels containing different types of fluorescent dyes and quantum dots. The composites demonstrated high stretching and healing properties. The obtained hydrogels were used for the creation of a LED device with high fluorescence emission. All these make the paper by Liu et al. interesting for a wide audience of the Nanomaterials journal.
The paper is well organized, logical, and provides full information about all stages of the work. Only a few recommendations can be done.
- Fluorescent hydrogels are quite been well studied last years and many papers are devoted to this topic. The authors give a nice review of them. However, a couple more papers worth to be mentioned: Adv. Funct. Mater., 20: 976-982 (https://doi.org/10.1002/adfm.200901812); Mater. Chem. , 2019,3, 1489-1502 (https://doi.org/10.1039/C9QM00127A); ACS Nano 2019, 13, 2, 1433–1442 (https://doi.org/10.1021/acsnano.8b07087).
- In the section "Materials and Methods", the protocol of the carbon quantum dots preparation is completely omitted. The authors just mentioned that it is described in their paper earlier. The authors should briefly describe this original protocol like it is done for the β-CD modification by MAH. This modification is also published elsewhere, so, why do not give more details for quantum dots?
- In the description of the frontal polymerization (p. 4, line 103), the authors wrote that one end of the horizontal vessel was heated to start the polymerization. However, the temperature is not pointed out.
- 4, lines 124-125. The authors do not need to repeat the typical composition of the gel. It is already described in section 2.3. The same is for the caption for Figure 1 (p. 5, lines 150-151).
- Figure 3e. The image clearly demonstrates the variation of the vibrations of the functional groups during the healing process. However, it is not clear which images correspond to the state “before” and “after”, what is the size of the mapped area, what was the time between these scans.
- Caption for Figure 5. The abbreviations TEOA, PPA, etc., and their explanations should be moved to the text.
- The paper contains very few grammar mistakes and misprints. The authors are recommended to carefully read the text once more.
The issues mentioned do not reduce the general high evaluation of the work, and the paper can be accepted to the Nanomaterials journal after minor revision.
Author Response
Comments to the Author
The paper under consideration reports the formation of self-healing hydrogels containing different types of fluorescent dyes and quantum dots. The composites demonstrated high stretching and healing properties. The obtained hydrogels were used for the creation of a LED device with high fluorescence emission. All these make the paper by Liu et al. interesting for a wide audience of the Nanomaterials journal.
The paper is well organized, logical, and provides full information about all stages of the work. Only a few recommendations can be done.
- Fluorescent hydrogels are quite been well studied last years and many papers are devoted to this topic. The authors give a nice review of them. However, a couple more papers worth to be mentioned: Adv. Funct. Mater., 20: 976-982 (https://doi.org/10.1002/adfm.200901812); Mater. Chem. , 2019,3, 1489-1502 (https://doi.org/10.1039/C9QM00127A); ACS Nano 2019, 13, 2, 1433–1442 (https://doi.org/10.1021/acsnano.8b07087).
Response: Done. Thanks for the reviewer’s comments. We have adopted the reviewer’s advice. Accordingly, we have cited relevant references and discussed in the introduction. The modified text is as follow:
Polymer gel provides an ideal matrix for hosting and immobilizing the QDs due to their unique porous structure. It should be noted that, the loading of QDs not only endows the composites of fluorescence, but also other enhanced fascinating features, such as self-healing [13]. Up to now, the QDs-loaded gels have found a wide scope towards sensors [5,6], wound dressing [7], drug delivery [8], photothermal therapy [9], optoelectronic devices [10], DNA detection [12], etc.
- In the section "Materials and Methods", the protocol of the carbon quantum dots preparation is completely omitted. The authors just mentioned that it is described in their paper earlier. The authors should briefly describe this original protocol like it is done for the β-CD modification by MAH. This modification is also published elsewhere, so, why do not give more details for quantum dots?
Response: Done. Thanks for the reviewer’s comments. We have adopted the reviewer’s suggestion. We have added details for the preparation of carbon quantum dots and CdTe QDs. The modified text is as follow:
2.2 Preparation of the CQDs
CQDs with bright green and blue fluorescence color were prepared according to our previse work [3]. Briefly, Fe3O4 (10 wt%), carbamide and zinc citrate dehydrate or sodium citrate dihydrate (carbamide: citrate = 5:1 mol/mol) were added into a flask, which was placed under an external magnetic field (450 kHz) for 3~5 min to excite the reaction to fabricate CQDs. The as-prepared CQDs were dispersed in water, followed by centrifugation at 10,000 rpm for 10 min to remove Fe3O4. Then the supernatant was further purified by ultra-filtration membrane (0.22 μm). After drying process, the solid samples were collected for further use.
2.3 Preparation of the CdTe QDs
CdTe QDs were synthesized by a modified procedure according to a literature [44]. 0.1136 g NaBH4 and 0.0638 g tellurium powder were added into a seed bottle, followed by injection of 2 mL pure water. The seed bottle was connected with a needle to release the as-produced H2. After reaction (~8 h), the solution becomes clear and NaHTe solution was obtained.
0.2284 g CdCl2·2.5H2O was dissolved in 40 g water in a flask. Then the ligand solution (0.2448 g NAC dissolving in 5 g H2O) was slowly added into the flask, followed by stirring to ensure full coordination of ligand and Cd2+. The pH value was adjusted to 9 by 5 M NaOH solution. After that, the as-prepared precursor solution was placed in an oil bath with N2 protection. When the temperature raised to 98 oC, the NaHTe solution was rapidly injected into the system. Green-color CdTe QDs were obtained after reaction for 1 h, which was precipitated with ethanol, centrifuged at 10000 rpm for 10 min, and then dried for further use.
- In the description of the frontal polymerization (p. 4, line 103), the authors wrote that one end of the horizontal vessel was heated to start the polymerization. However, the temperature is not pointed out.
Response: Done. Good suggestion. We have adopted the reviewer’s comments and added the initiating temperature. The modified text is as follow:
The initiating temperature was 100 oC and the distance from the liquid level was 10 mm.
- lines 124-125. The authors do not need to repeat the typical composition of the gel. It is already described in section 2.3. The same is for the caption for Figure 1 (p. 5, lines 150-151).
Response: Done. Thanks for the reviewer’s comments. According to the reviewer’s suggestion, we have deleted the repeated descriptions. The modified text is as follow:
Typically, we synthesized poly(AM-co-AMPS-co-MAH-β-CD) gels via horizontal FP. To visually observe the propagating front, methylene blue was utilized, which will decompose when heated.
Figure 1. (a) Photographs demonstrate the propagating process of the front. (b) Frontal position-time curve in a typical FP process. Corresponding (c) IR thermal images and (d) temperature profile. (e) Frontal position-time curves under different AM/AMPS weight ratios. (f) Frontal velocity and Tmax under different AM/AMPS weight ratios.
- Figure 3e. The image clearly demonstrates the variation of the vibrations of the functional groups during the healing process. However, it is not clear which images correspond to the state “before” and “after”, what is the size of the mapped area, what was the time between these scans.
Response: Done. Good suggestion. We have adopted the reviewer’s comments. Accordingly, we have demonstrated the “before” and “after” healing process in the image with “original”, “cut” and “healed”. The scale bar is marked in the image. Details about the Micro-IR imaging measurements are added.
The modified text and figure are as follow:
Micro-IR imaging measurements. IR images during the self-healing process was recorded by a Thermo Scientific Nicolet iN10 infrared microscope, equipping with a liquid nitrogen cooled MCT detector (Thermo Electron Corporation, USA). The data were collected under reflection mode. IR images were captured using an aperture size of 50 μm by 50 μm. These data were abalysed by OMNIC picta software (Thermo Electron Corporation, USA).
Figure 3. (a) Photograghs show the self-healing properties of poly(AM-co-AMPS-co-MAH-β-CD) gels. (b) Self-healing efficiency versus different AM/AMPS mass ratios. (c) Schematic illustrating the healing mechnism. (d) FT-IR spectra of the gel. (e) Micro-IR images of gels before and after self-healing.The scale bar is 10 μm.
- Caption for Figure 5. The abbreviations TEOA, PPA, etc., and their explanations should be moved to the text.
Response: Done. Thanks for the reviewer’s comments. We have adopted the reviewer’s advices. The explanations of the abbreviations have been moved to the main text. The modified text is as follow:
As shown in Figure 5b, c, after the upper layer contacting with TEOA (triethanolamine), PPA (n-propylamine), TEA (triethylamine), TMEDA and EDA (ethylenediamine), the bilayer-structure exhibits varied luminescence intensity.
Figure 5. (a) Schematic illustration of a fluorescent dual-signal gels assemblies for 2D codes application. (b) Fluorescence spectra of the gel assemblies toward different organic amines (TEOA, PPA, TEA, TMEDA, and EDA).
- The paper contains very few grammar mistakes and misprints. The authors are recommended to carefully read the text once more.
Response: Done. Thanks for the reviewer’s comments. We have carefully revised the grammar mistakes and misprints.

Round 2
Reviewer 1 Report
All the comments from the reviewers were addressed by the authors. The overall quality of the paper has improved and reach publication level.
Reviewer 2 Report
The authors have taken into account all comments by the reviewer and essentially improved the text and the presentation of the results. The paper can be accepted in the present form.